# Evaluation of Shoulder Microcirculation Abnormality Using Laser Doppler Flowmetry

**DOI:** 10.3390/diagnostics12010143

**Published:** 2022-01-07

**Authors:** Taipau Chia, Jian-Guo Bau, Guo-Dung Hung, Sz-Huan Tsai, Che-Ming Hu

**Affiliations:** 1Department of Safety, Health and Environmental Engineering, Hungkuang University, Taichung City 433, Taiwan; tpchia38@gmail.com (T.C.); sean811002@gmail.com (S.-H.T.); 2Department of Biomedical Engineering, Hungkuang University, Taichung City 433, Taiwan; 3Department of Rheumatology and Immunology, Kuang Tien General Hospital, Taichung City 433, Taiwan; 4Safety & Health Section, YC INOX Co., Changhua 524, Taiwan; cheming1981@msn.com

**Keywords:** neck-shoulder pain, microcirculatory blood flow, Laser Doppler Flowmetry

## Abstract

Severe neck-shoulder pain induces functional limitations in both life and work. The purpose of this study was to determine the characteristics of shoulder microcirculation abnormality in workers. This study recruited 32 workers and patients, both *n* = 16. Questionnaires were administered, and Laser Doppler Flowmetry (LDF) was used to measure microcirculatory blood flow (MBF) at the myofascial trigger points (MTrPs) on the shoulders. The absolute-deviation_MMBF_ represented the mean MBF (MMBF) variability among subjects. The differences in the life characteristics, shoulder pain level, and microcirculatory characteristics at MTrPs between the two groups were compared. It was found that shoulder pain level was significantly higher in the patient than in the control group (*p* < 0.001). Deviation of the MMBF value beyond the postulated “normal range” of 60–80 was significantly higher in the patient than in the control group (*p* < 0.001). The MMBF deviation was significantly correlated with shoulder pain level, pain duration, and the symptom effect (*p* < 0.01, *n* = 32). A normal range for the MMBF of 60–80 on the shoulder near MTrPs is hypothesized for the first time based on this study. Noninvasive LDF can be used to assess abnormality in the MBF on shoulder MTrPs at an early stage.

## 1. Introduction

In the 3C-prevalent age, most people have become physically inactive. Physical inactivity due to societal changes toward more sedentary behaviors is leading to health problems [1]. In particular, severe neck-shoulder pain (NSP) caused by sedentary life has affected the quality of both work and life. Risk factors for developing neck pain include low satisfaction with the workplace environment, keyboard position close to the body, low work-task variation, and self-perceived medium/high muscular tension [2]. Factors related to work organization and psychosocial factors also have indirect impacts on the risk of chronic shoulder pain [3]. 

Health problems are emerging among workers in Taiwan due to unhealthy lifestyles and insufficient physical activity. Three centers for promoting workplace health have been established since 2003, as well as nationwide services including an injury reporting system, health-promotion program consultations, train-the-trainer courses, and other educational tools [4]. Furthermore, the continuous development of workplace health-promotion measurement tools is the focus of the next phase of efforts aimed to addressing the severity of musculoskeletal disorders. Methods to assess shoulder and neck pain include the visual analogue scale (VAS) score [5], Neck Disability Index, neck mobility, muscle strength [6,7], white-light spectroscopy, Laser Doppler Flowmetry (LDF) [8,9], and quantitative imaging biomarkers [10]. Salgado [11] discussed several techniques for evaluating microvasculature perfusion, such as LDF, intravital microscopy, and orthogonal polarization spectral imaging techniques. The capillary refill time constitutes an accessible route for probing microcirculation, which is applied during shock resuscitation [12]. Valid and reliable biomarkers could be used to objectively evaluate the effects of medical treatments and workplace interventions [10]. Pan et al. suggested that the skin perfusion pressure (SPP) measured using LDF may be applied as an index for accurately predicting wound healing in patients with limb ischemia [13]. The consensus from previous studies is that noninvasive measuring techniques have been widely accepted.

Our previous study using LDF found that the mean blood flow in the neck/shoulder region was significantly lower in the high-pain group than in the low-pain group [14]. This demonstrated that the mean microcirculatory blood flow (MMBF) of the shoulder region was affected by aging and had lower values at higher pain levels. Furthermore, our previous study investigating shoulder microcirculation using LDF indicated that even with a low exercise volume, people with regular physical activity had noticeable differences in their microcirculatory characteristics [7], with high blood flow only being observed in people in the study who did not exercise at all. Lin illustrated that LDF indices can be used to evaluate blood-flow-perfusion responses and their regulation [15]. Jacob concluded that it is high time to give more attention to bedside techniques providing insight into local tissue perfusion [16].

The purpose of the present study was to determine the characteristics of shoulder microcirculation among patients and controls using noninvasive LDF. The MMBF was measured at the myofascial trigger points (MTrPs) on the shoulder, where myofascial pain usually occurred. The differences of age, shoulder pain level, and MMBF between patients and controls, and the correlations of these variables, were investigated. It is well known that the blood pressure of a healthy person should fall within a normal range. We therefore also investigated whether there is a normal range for the skin microcirculation on shoulder MTrPs of asymptomatic subjects.

## 2. Materials and Methods

### 2.1. Subjects

The present study recruited 32 employees, who provided informed consent. Each employee was from one hospital, one university, or one high-technology factory. Sixteen employees with nontraumatic neck and shoulder pain were assigned to the patient group (age = 40.6 ± 6.5 years, mean ± SD) when they first visited their rehabilitation doctor. The remaining 16 were assigned to the control group (age = 35.3 ± 4.3 years). Those from the university and factory assigned to the control group had not yet visited a rehabilitation doctor at a hospital. The numbers of males and females in the patient and the control groups were male (*n* = 4), female (*n* = 12) and male (*n* = 7), female (*n* = 9), respectively.

### 2.2. Measurements

Questionnaires were administered to investigate personal social variables, self-perceived pain levels, and work and life conditions. A VAS with levels ranging from 1 to 10 was used to evaluate pain levels. The duration of pain was categorized into six levels: (1) 1 month, (2) 3 months, (3) 6 months, (4) 1 year, (5) 3 years, and (6) more than 3 years. The symptom effect was also categorized into six levels: (1) no impact on life and work activities, (2) slightly lower work ability, (3) obviously lower work ability, (4) has taken days off work for recovery, (5) reduced ability to perform the activities of daily living, and (6) inability to move at all.

The skin blood perfusion of the patient group was measured before treatment during their first visit to the rehabilitation department. Since myofascial pain can be diagnosed by the presence of one or more myofascial trigger points (MTrPs), it was defined as a hyperirritable spot in a palpable taut band of skeletal muscle fibers [8,17,18]. A high-power LDF device (VMS-LDF1-HP, Moor Instruments, Axminster, Devon, UK) operating at 785 nm and a laser power of less than 20 mW (complying with Class 3R of the IEC 60825-1:2007), and with a wide-separation (4 mm) noninvasive skin probe (VP1-V2-HP), was used to measure the microcirculatory blood perfusion at MTrPs on the affected shoulder for the patient group. The probe was placed at a fixed position on point a or b illustrated in Figure 1. Blood perfusion at the shoulder of subjects with a lower pain level (mostly with no pain) was measured for the control group (point a or b). Point C7 is the seventh cervical vertebrae. The distance between points a and C7 or points b and C7 was 45% of the distance from C7 to the acromion (point S_1_ or S_2_). Microcirculatory perfusion signals of each subject were recorded for 5 min. Blood perfusion was then obtained by analyzing the characteristics of the refracted light and was expressed in arbitrary units defined by the ratio of the measured light intensity to the light intensity obtained in an aqueous suspension of polystyrene latex particles during calibration. MMBF represents the mean blood flow of each subject during the 5 min measurement. The detailed working theory and algorithms of LDF were described previously [19], and other measurement conditions and requirements such as LDF recording duration and room temperature were based on our other previous studies [9,14].

### 2.3. Data Analysis

To understand the variability of MMBF among the subjects, the mean absolute deviation of MMBF, denoted as absolute-deviation_MMBF_, was calculated using the following formula of absolute-deviation Equation (1):(1)$${\text{absolute-deviation}}_{\mathrm{MMBF}}=\left[\sum \left|x_{i}-\mu \right|\right]/n$$
where *µ* represents the mean MMBF of the group, $x_{i}$ represents the MMBF of each subject in the group, | | represents the absolute value, and *n* represents the number of subjects.

According to the studies of Castronuovo [20] and Pitts et al. [21], the normal SPP range for a healthy adult is 50–100 mmHg. Our previous research indicated that the MMBF values of subjects with NSP were typically <60 or >80, and hence deviating from the range of “60–80”. Therefore, this study postulated that a normal range of MMBF in the shoulder around the frequently occurred sites of MTrPs is 60–80, and so the limits of 60 and 80 were selected for convenience. The definition for deviations from this normal range is provided below. MMBF values inside the normal “60–80” range were not included when calculating the deviation.

MMBF deviation beyond the normal range =
|MMBF (>80) − 80| or |MMBF (<60) − 60|(2)

### 2.4. Statistical Analysis

Statistical analysis was performed using SPSS (version 26.0, SPSS Incorporated, Chicago, IL, USA). Differences in the characteristics and deviations of the shoulder MMBF between the two study groups were analyzed using the Mann–Whitney U test. Correlations between MMBF with age and perceived pain symptoms were analyzed using Spearman analysis. Since the study included two independent groups with limited samples, a two-tailed significance level of 5% was adopted.

## 3. Results

Table 1 compares the physical parameters and lifestyles of the two groups. Age and perceived shoulder pain level were both significantly higher in the patient group (*p* < 0.05 and <0.001, respectively). The control group spent more time using 3C products (*p* = 0.048). The patient group had marginally lower MMBF than the control group (*p* = 0.065), but the absolute-deviation_MMBF_ of the patient group was significantly higher (*p* < 0.001). The mean pain level of the patient group for different genders was 6.0 for male (*n* = 4) and 6.6 for female (*n* = 12). The mean pain level of the control group was 0.2 for male (*n* = 7) and 1.9 for female (*n* = 9). There was no significant difference in pain level between different genders.

Table 2 lists the relationships between age, MMBF, MMBF deviation, shoulder pain level, pain duration, and symptom effect for all the subjects (*n* = 32). Shoulder pain level, pain duration, and MMBF deviation were all significantly correlated with age (*p* < 0.05). MMBF was negatively correlated with MMBF deviation, while MMBF deviation was significantly correlated with shoulder pain level, pain duration, and symptom effect. Shoulder pain level was significantly correlated with pain duration and symptom effect, and pain duration was significantly correlated with symptom effect.

Figure 2 shows the distributions of pain level and shoulder MMBF in the two groups. The MMBFs of the control group (*n* = 16) are presented in the left part of the figure, showing who had increasing pain levels. The right part shows that the pain level of the patient group (*n* = 16) gradually increased up to its highest level of 10. When MMBF values were similar, absolute-deviation_MMBF_ differed significantly between the two groups (Table 1).

Based on our assumption related to the normal range, the difference of MMBF deviating beyond the normal range between the two groups was investigated. Figure 3 compares shoulder pain level, MMBF, and deviation beyond the normal range between the patient and control groups. The results indicate that the MMBF of the two groups were similar, but that the deviation beyond the normal range was markedly higher in the patient group (19.5 ± 11.4) than in the control group (2.2 ± 2.7, *p* < 0.001).

## 4. Discussion

This study used noninvasive LDF to investigate the MMBF at MTrPs for subjects with different shoulder pain levels. A significant difference was observed in the deviation from the normal MMBF range between the patient and control groups. In the questionnaire survey, the percentages of subjects who reported that their shoulder pain partly or fully affected their jobs were 31% and 75% in the control and patient groups, respectively.

Regarding the assumption of the normal MMBF range of 60–80, since the averages of the two groups were similar but the deviation was larger in the patient group, it is speculated that the normal value for shoulder perfusion is between 60 and 80. The lower and upper limits of 60 and 80 were assumed based on empirical results from our previous research, in which the shoulder perfusions of more than 50 healthy subjects were measured. Their MMBF range was indicated to be approximately 60–80 (unpublished data). Moreover, the normal systolic and diastolic blood pressures of healthy people are considered to be 110–130 mmHg and 70–90 mmHg, respectively. In the far end of blood circulation, the final goal of microvascular blood flow per unit of time is to ensure the needed exchange of substances between tissue and blood compartments [16]. There should also be a range for terminal microcirculation. Therefore, the normal MMBF range of 60–80 at the shoulder being near to frequent MTrP sites was postulated in the present study.

This study analyzed the microcirculation of those in the patient group who experienced shoulder pain to understand the mechanism of pain at shoulder MTrPs. The formation of MTrPs, which is now widely accepted, is the “integrated hypothesis of energy crisis” proposed by Travell and Simmons [17,22,23]. This energy crisis is initiated through MTrPs that are defined as hyperirritable spots within taut bands of skeletal muscle fibers, and it is caused by an excessive load on the muscles that leads to dysfunction of the motor endplate, resulting in uncontrolled shortening of muscle fibers (taut band) and continuous contraction of the sarcomere (contraction knot). As sarcomere contraction occurs, local metabolic demand increases. The diminishing of local blood circulation results in local hypoxemia, leading to the energy crisis and pain [23]. Our previous study found that the shoulder MMBF was significantly lower in the high-pain group than in the low-pain group (*p* < 0.05) [14]. Reduction in microcirculatory perfusion in MTrPs reflects the pain symptom of the worker.

Figure 2 presents the lower perfusion of 11 patients in the patient group (MMBF <60 on the right side). The MMBF of the patient with the highest pain level (10) was 32.9. Consistent with other studies, neck muscles might become smaller in the presence of neck pain, and pain in the trapezius myalgia and neck/shoulder may be associated with reduced blood flow, relative blood volume, and oxygen saturation at rest [10,24]. Takiguchi [23,25] found that a biomarker of trapezius metabolism in neck/shoulder pain was inversely correlated with pain, indicating that muscle metabolism reduced in this condition. Gold [10] concluded that reductions in blood flow and oxygen saturation may facilitate the production of muscle metabolites such as lactate, which are known to influence muscle nociceptor activity.

Using a microanalytic technique, Shah [26] found that subjects with active MTrPs in the trapezius muscle have a biochemical milieu of selected inflammatory mediators, neuropeptides, cytokines, and catecholamines different from subjects with latent or absent MTrPs in their trapezius. Pober [27] suggested that during acute inflammatory responses, leukocyte-derived mediators such as histamine and bradykinin will cause arteriolar dilation, thereby increasing blood flow leading to rubor (redness and swelling). If the stimulus persists, inflammation evolves into a chronic phase. Inflammation has been suggested to occur in early stages of the disease, whereas fibrosis, as a result of collagen and matrix synthesis, occurs in a later stage [28,29]. There is evidence that the transition from acute to chronic inflammation relies on an angiogenic response as a means to provide blood supply to inflamed neotissue [27]. In our study, there were four subjects in the patient group with a high pain level (range 6–8), whose MMBF values also appeared to be abnormal (>80; Figure 2). MTrPs in the upper trapezius muscle of these subjects were speculated to occur in rubor and inflammatory status.

One particularly interesting result of the present study was that the average blood pressures in the two groups were similar and within the normal range. However, the shoulder MMBF in MTrPs of the patient group tended to deviate from the normal 60–80 range. Measurements of mean microcirculatory blood flow (MMBF) at MTrPs indicated that some patients had blood insufficiency (MMBF <60) or excessive blood perfusion (MMBF >80). In the statistical analysis of the degree of deviation from the normal 60–80 range, the patient group deviated significantly more than the control group (*p* < 0.001; Figure 3). Meanwhile, there was no obvious difference observed between the MMBF in MTrPs of the two groups. A study related to sepsis found that microcirculatory alterations can be detected in very early stages of the disease and may persist regardless of the macrohemodynamic status [11,30,31]. Those findings were consistent with our observations. Furthermore, one can see inflamed synovia with neoangiogenesis (related to inflammation) and thickened and stiffened joint capsule (related to fibrosis) during surgery [32]. We believe that the time course of shoulder tissue will elucidate signs of abnormal microcirculation at early stages of pain development. However, the inflammatory reactions or ischemia that evolves and differentiates at MTrPs needs further investigation of more subjects with different time courses of pain.

All subjects in this study were 30–45 years old, and there was a significant difference between the ages in the two groups; the control group was 5 years younger than the patient group on average and had used 3C products for longer periods (Table 1). The subjects in the control group had not yet reached the point of visiting rehabilitation physicians, and it was inevitable that some subjects had a pain level of 4 on the VAS due to their sedentary lifestyles. A prolonged stationary posture may cause muscle fiber damage and interfere with Ca^+2^ homeostasis, resulting in muscle pain [23], and the neck muscle size may be decreased with pain. Additionally, trapezius myalgia and neck/shoulder pain may be associated with reduced vascularity in the trapezius and reduced trapezius oxygen saturation at rest and in response to upper extremity tasks [10]. With the impairments that accompany aging, shoulder pain level, pain duration, and MMBF deviation significantly increased (Table 2), which seems to indicate that the subjects in the control group may eventually need to see a doctor because of worsening pain if they still maintain an inactive lifestyle when they reach 40 years old. Hallman [33] found an association between sitting time (in total per day and specifically during work) and NSP intensity among blue-collar workers. For the causes of NSP, both occupational and nonoccupational factors could be associated with the onset of shoulder pain among adults aged 40 years and older [34]. Higher NSP intensity and higher physical work demands, and particularly their combination, were associated with higher odds of work limitation due to pain among older workers [35]. The International Organization for Standardization 45,001 and Occupational Health and Safety Management System have provided guidance for organizations to implement health monitoring of employees, if feasible, through appropriate medical supervision or tracking signs and symptoms that are harmful to the health of employees at an early stage to ensure the effectiveness of prevention and control measures. It is necessary to treat the triggering factors, such as improving working posture, encouraging moderate stretching and rest, and reducing workload [36]. Conclusively, Jacob [16] mentioned that only widespread insight into microvascular physiology and pathophysiology has the power to improve diagnostics, leading to a real target-oriented therapy of oxygen delivery in critically ill patients. The results of the present study might also provide evidence-based support for monitoring subjects with shoulder myofascial pain early.

The limitations of this study included cross-sectional design and insufficient subject samples in both groups. Although the normal range of 60–80 was in arbitrary unit based on the definition used in the calibration process, the present study demonstrated the existence of this normal range. On the other hand, the normal range should decrease with age, and an exact normal range should be confirmed using more subjects with a different age range. The present study indicated a significantly larger deviation of MMBF in the patient group, which provided support for the development of microcirculatory abnormality at MTrPs. The other limitation is no diagnostic result of the patient group regarding the anatomical location of the pain. According to our recent research [37], we also found that the variability of MMBF among the patients with higher Neck Disability Index and limited cervical range of motion was significantly high, even though the difference of MMBF between the asymptomatic subjects and the patients was nonsignificant. Therefore, future study of myofascial pain needs to combine biochemical tests, physical examinations, and blood flow measurements to provide physicians a better understanding of the pathogenesis and mechanism of neck/shoulder pain. Moreover, it would be interesting to conduct further study into the gender differences in MBF for more subjects, as limited information was provided for this study.

## 5. Conclusions

This study used LDF to investigate the MMBF at MTrPs of the subjects with various levels of shoulder pain, and it is the first to postulate the normal range of MMBF at the shoulder sites near where MTrPs frequently occurred. A significant difference of MMBF deviating from the 60–80 range was observed among those in the patient group with severe shoulder pain. Patients with MMBF >80 at the shoulder may be in a reactive hyperemia, rubor, or inflammatory state, while those with MMBF <60 may be in a local hypoxemia or tissue ischemia state. With the impairments that accompany aging, shoulder pain level, pain duration, and the MMBF, deviation increased significantly in the patient group. These results demonstrated that noninvasive LDF has the capability of monitoring MMBF abnormality at shoulder MTrPs at an early stage. It is suggested that the MMBF value may be useful for indicating pathogenic processes or responses to rehabilitation treatment. Further study is needed to confirm the microcirculatory characteristics of patients with shoulder pain in different phases of its pathology based on clinical evidence.

## Figures and Tables

**Figure 1 diagnostics-12-00143-f001:**
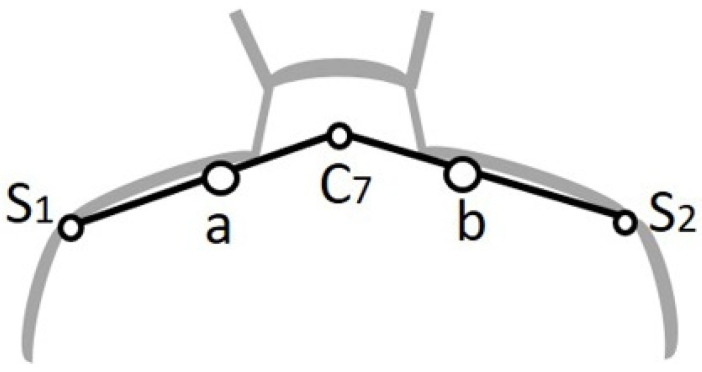
Diagram of the LDF measurement to evaluate shoulder microcirculatory blood flow at point a or b. C7: seventh cervical vertebrae. S_1_ and S_2_: acromion of shoulders.

**Figure 2 diagnostics-12-00143-f002:**
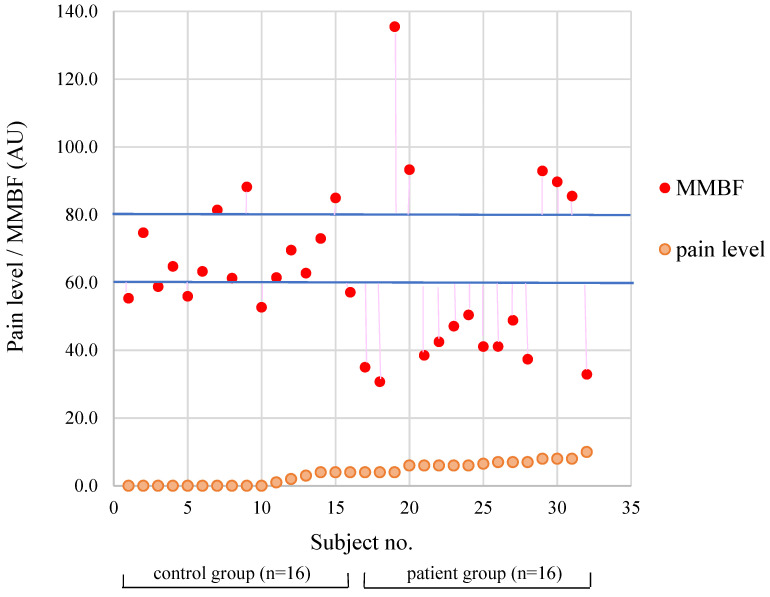
Distributions of pain level and shoulder mean microcirculatory blood flow (MMBF) in the subjects (*n* = 32).

**Figure 3 diagnostics-12-00143-f003:**
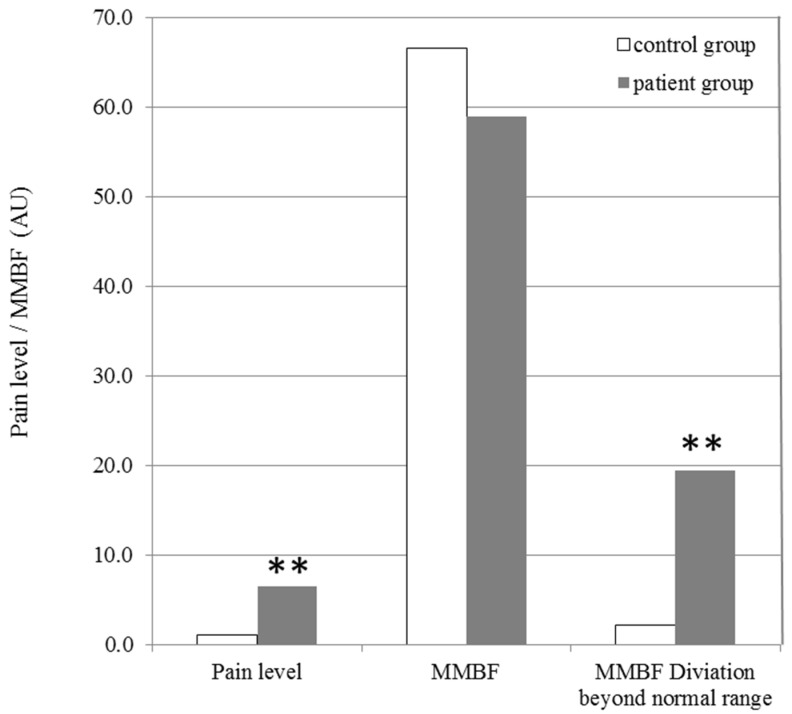
Comparisons of the shoulder pain level, MMBF, and deviation from the normal range (>80 or <60) between the patient and control groups (** *p* < 0.001).

**Table 1 diagnostics-12-00143-t001:** Differences in physiological parameters, life characteristics, shoulder pain level, and mean microcirculation blood flow (MMBF) between the patient and control groups.

	Control Group (*n* = 16)	Patient Group (*n* = 16)	*p* Value
Age (years)	35.3 ± 4.3	40.6 ± 6.5	0.016 *
BMI (kg/m^2^)	22.2 ± 2.8	24.4 ± 3.7	0.09
SBP (mmHg)	117.6 ± 11.9	122.7 ± 16.3	0.25
DBP (mmHg)	73.5 ± 9.8	74.3 ± 10.5	0.925
Exercise/week (hours)	2.4 ± 3.1	0.9 ± 1.7	0.119
Sitting/day (hours)	6.4 ± 1.7	6.3 ± 3.2	0.499
Time using 3C products/day (hours)	7.5 ± 2.9	4.8 ± 3.3	0.048 *
Shoulder pain level	1.1 ± 1.6	6.5 ± 1.6	<0.001 **
MMBF	66.6 ± 10.6	58.9 ± 29.6	0.065
Absolute-deviation_MMBF_	9.0 ± 5.6	9.9 ± 15.3	<0.001 **

Data are mean ± SD values analyzed using the Mann–Whitney U test; *: *p* < 0.05; **: *p* < 0.01.

**Table 2 diagnostics-12-00143-t002:** Relationships between age, shoulder pain parameters, MMBF, and MMBF deviation for all participants (*n* = 32) ^θ^.

	Age	MMBF	MMBF deviation	Shoulder Pain Level	Pain Duration	Symptom Effect
Age (years)	1					
MMBF	−0.066	1				
MMBF deviation	0.350 *	−0.432 *	1			
Shoulder pain level	0.408 *	−0.202	0.650 **	1		
Pain duration	0.392 *	−0.230	0.641 **	0.819 **	1	
Symptom effect	0.262	−0.227	0.450 **	0.792 **	0.662 **	1

*: *p* < 0.05; **: *p* < 0.01; ^θ^: analyzed using Spearman’s *ρ* test.

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
