# Peer review of "Evaluation of Shoulder Microcirculation Abnormality Using Laser Doppler Flowmetry"

_diagnostics, 2022, doi:10.3390/diagnostics12010143_

Round 1

Reviewer 1 Report

It is a quite interesting manuscript. The topic of this manuscript falls within the scope of Diagnostic Journal.

The purpose of this study was to determine the characteristics of shoulder microcirculation abnormalities in workers. This study recruited 32 workers and patients both n = 16.  The Authors used questionnaires a Laser Doppler Flowmetry (LDF) to measure microcirculatory blood flow (MBF) at the myofascial trigger points (MTrPs) on the shoulders.  The absolute deviation of MMBF was significantly correlated with shoulder pain level, pain duration, and the symptom effect.

On the basis of obtained results the Authors conluded that noninvasive Laser Doppler Flowmetry may be used in monitoring MBF abnormalities at shoulder MTrPs at an early stage as well as the MBF value may be useful for indicating pathogenic processes or responses to rehabilitation treatment.

The  data has been provided with vigorous statistical analysis. The Authors have presented sufficient data. The appropriate tables and figures have been provided. The manuscript is well written. The article is easy to read and logically structured.  The methods are adequately described.  The Authors also added very good subchapter in discussion- limitations. The conclusions are consistent with presented evidence and arguments. They address the main question posed.

Author Response

Thanks for your valuable review comments. We are very happy to be affirmed.

Reviewer 2 Report

Chia et al. present a somewhat preliminary study on shoulder microcirculation evaluation using Laser Doppler Flowmetry LDF. They conclude that LDF can be used to access abnormalities on shoulder myofascial trigger points at early stages. While the scope is very limited and amount of new information is relatively small, the study is well-written and relatively well-presented, without any major flaws. Suggestions for improvement as listed below.

Abstract

Please define “3C”, “LDF” and “MTrPs” (better: do not use abbreviation if possible since it is used only once and use of abbreviations should be reduced in abstracts)

Introduction

Please define “3C”

Title

It is unclear which “abnormalities” (as stated in the title) were evaluated. Examining the abstract and the content of the study it seems that only one abnormality is being evaluated and reported: the deviation of MMBF. (The first line of Discussion confirms this.) As a suggestion, the singular form may be used (“abnormality”) in the title or “abnormalities” may be removed.

Introduction

The sentence “The differences between and correlations of MBF in the shoulder with pain levels between patients and controls were investigated.” should be revised for clarity. It is unclear what is intented by “differences between”.  Differences between MBF and pain level? How these could even be compared using the same parameters?

The expression “normal range for the skin microcirculation” should also be revised. It is understandable to expect a normal range for each physiological parameter in the skin microcirculation. However, normal range for the whole skin microcirculation would certainly involve a large number of factors.

The concept of myofascial trigger points should be briefly presented at Introduction and so its abbreviation (MTrPs).

Materials and Methods

2.2. Measurements, last sentence – Although the text cites reference 14 for the LDF measurements, not all details are included in that study. Please add reference 9 (Bau et al. 2014) to the text, where more details are explicitly described. It should be mentioned also that other pertinent information such as LDF recording duration and room temperature are the same as in reference 14.

Discussion

The first statement of the third paragraph is questionable: “This study further analyzed microcirculation of those in the patient group who experienced shoulder pain to understand the mechanism of pain at shoulder MTrPs.” The authors should make it clearer in the manuscript text how the microcirculation was “further analyzed” since no other measurement (besides MMBF deviation) is reported.

Table 1 – Please add the gender information (number/%, males / females) of control and patient groups. Please indicate in the manuscript text whether results were different between male and female subjects.

Figure 1 and respective legend – Please indicate the meaning of the labels “a”, “b”, “S1”, “C”, “S2”, “PC” and “Moor LDF”. Please indicate also the meaning and the difference between the straight blue lines and curved black lines.

Figure 3 and respective legend – Please add a y-axis title (include units). Please also indicate the meaning of “**”.

Reviewer 3 Report

Major concerns,

This certainly is an interesting observatory study. My major concern is that as the LDF measurement is superficial, there's a lack of strong evidence from the conventional clinical data supporting the correlation between the deviation of the MMBF and the patients' symptoms.  Since there's no diagnostic result of the patient group regarding the anatomical location of the pain, whether it's acute or chronic. It makes the conclusion from LDF less scientifically sounding.

Minor concerns,

1) Please clarify the meaning of "3C".

2) Page 7: missing a period in the second last paragraph.

3) Fig 2: It would be helpful if the pain level is shown in a secondary y-axis, and two groups are clearly labeled.

Round 2

Reviewer 3 Report

I appreciate the authors' address to my previous concerns. My major beef with this manuscript is still this correlation between pain level with the deviation from an arbitrarily chosen "normal range".

1) Let's say MMBF is a good indication of increased/decreased local blood flow. Then physiologically speaking, how would MMBF be upregulated by no workout (and quickly downregulated by quick stretching) (https://doi.org/10.1007/s40846-017-0248-y) while downregulated by higher pain level (https://doi.org/10.1371/journal.pone.0169318)? 

2) Besides, why previous method (PP value) was not applied in this manuscript, if the MMBF value was considered not accurate enough previously? (https://doi.org/10.1371/journal.pone.0169318)

3) What are the criteria of this normal range of MMBF? Is this range derived from a large sample size or an alternative method?

I feel like there's still some work that needs to be done in the introduction and discussion part, so the relevence of this non-invasive evaluation of microcirculation can be linked to tissue damage (neck-shoulder pain). Again I have no issue with the current data but need more biological data (the authors or the others' work) to be convinced by the story.

Minor concerns,

Page 2 line 62: The previous study done by the authors' group should be ref 8, not 7.

Fig 2: Pian level is still hard to read.